# Inhibition of phosphatidylinositol 3-kinase α (PI3Kα) prevents heterotopic ossification

José Antonio Valer[1], Cristina Sánchez-de-Diego[1], Beatriz Gámez[1], Yuji Mishina[2], José Luis Rosa[1] & Francesc Ventura[1,*] (iD)

## Abstract

Heterotopic ossification (HO) is the pathological formation of ectopic endochondral bone within soft tissues. HO occurs following mechanical trauma, burns, or congenitally in patients suffering from fibrodysplasia ossificans progressiva (FOP). FOP patients carry a conserved mutation in *ACVR1* that becomes neomorphic for activin A responses. Here, we demonstrate the efficacy of BYL719, a PI3Kα inhibitor, in preventing HO in mice. We found that PI3Kα inhibitors reduce SMAD, AKT, and mTOR/S6K activities. Inhibition of PI3Kα also impairs skeletogenic responsiveness to BMPs and the acquired response to activin A of the *Acvr1*$^{R206H}$ allele. Further, the efficacy of PI3Kα inhibitors was evaluated in transgenic mice expressing *Acvr1*$^{Q207D}$. Mice treated daily or intermittently with BYL719 did not show ectopic bone or cartilage formation. Furthermore, the intermittent treatment with BYL719 was not associated with any substantial side effects. Therefore, this work provides evidence supporting PI3Kα inhibition as a therapeutic strategy for HO.

**Keywords** bone; bone morphogenetic protein; fibrodysplasia ossificans progressiva; heterotopic ossification; PI3K
**Subject Categories** Chemical Biology; Genetics, Gene Therapy & Genetic Disease; Musculoskeletal System

## Introduction

Heterotopic ossification (HO) is a pathology characterized by ectopic bone formation in soft tissues at extraskeletal sites. Trauma-induced HO develops as a common post-operative complication after orthopedic surgeries (e.g., hip arthroplasty), blast injuries, skeletal trauma, deep burns, and nervous system injuries. Clinical therapy is now limited to anti-inflammatory drugs, radiation, or surgical excision of the already formed bone, which is associated with a higher recurrence rate (Dey *et al*, 2017; Eisenstein *et al*, 2017). In addition to trauma-induced HO, fibrodysplasia ossificans progressiva (FOP)

is a devastating congenital autosomal dominant disorder also involving HO (Bravenboer *et al*, 2015). Ectopic bones form progressively through endochondral ossification, mostly in episodic flare-ups associated with trauma and inflammation. Cumulative effects of these osteogenic events include progressive immobility resulting from ankylosing joints. Individuals affected have a shorter life span, most commonly due to thoracic insufficiency syndrome (Kaplan *et al*, 2010; Pignolo *et al*, 2016).

The genetic cause of FOP arises from gain-of-function mutations in the bone morphogenetic protein (BMP) type I receptor *ACVR1* (Shore *et al*, 2006). Several reports demonstrated that *ACVR1* mutations in FOP are neomorphic, abnormally transducing BMP signals in response to activin A (Hatsell *et al*, 2015; Hino *et al*, 2015). Therefore, whereas activin A normally transduces a pSMAD2/3 signal through the ACVR2-ACVR1B complex, it forms a non-signaling complex with ACVR2 and wild-type ACVR1. However, the neomorphic *ACVR1* allele in FOP elicits SMAD1/5 phosphorylation upon activin A binding to the ACVR2-ACVR1$^{R206H}$ receptor complex (Hatsell *et al*, 2015; Hino *et al*, 2015). In contrast to FOP, which results from improper activation of FOP-mutant ACVR1 by activin A, non-genetically driven HO appears to arise from excessive BMP signaling with functional redundancy between different BMP type I receptors (Agarwal *et al*, 2017).

Trauma-induced HO and FOP are initiated by local connective tissue destruction, and require inflammation (Pignolo *et al*, 2013; Convente *et al*, 2018). This inflammatory microenvironment activates a resident pool of interstitial progenitors that aberrantly undergo chondrogenesis and further ectopic bone formation (Dey *et al*, 2016; Lees-Shepard *et al*, 2018). BMP signaling plays a major inductive role in HO progression: chondrogenesis and osteoblast specification and maturation (Salazar *et al*, 2016). Therefore, targeting BMP signaling in all these cell types and pathogenic steps would be required for efficient therapeutic treatments.

Genetic and pharmacological studies indicated that chondroblast and osteoblast specification and maturation depend on phosphatidylinositol 3-kinase (PI3K; Fujita *et al*, 2004; Ford-Hutchinson *et al*, 2007; Ikegami *et al*, 2011). We have previously demonstrated that PI3Kα is critical for bone formation through regulation of SMAD1 degradation and activity (Gámez *et al*, 2016). The PI3K-AKT axis also integrates additional major osteogenic signaling

---

1 Departament de Ciències Fisiològiques, Universitat de Barcelona, IDIBELL, Hospitalet de Llobregat, Spain
2 Department of Biologic and Materials Sciences, School of Dentistry, University of Michigan, Ann Arbor, MI, USA
*Corresponding author. Tel: +34-934024281; Fax: +34-934024268; E-mail: fventura@ub.edu

pathways (e.g., mTOR, FOXO, or GSK3; Vanhaesebroeck *et al*, 2010; McGonnell *et al*, 2012). Recent studies also pointed out the critical role of mTOR in HO and FOP (Hino *et al*, 2017; Qureshi *et al*, 2017). Hence, we hypothesized that a therapy based on PI3K inhibitors could be effective because it would target both the chondrogenic and osteogenic steps of heterotopic ossification, by targeting multiple pathways downstream of PI3K/AKT. Among PI3K inhibitors, BYL719 is an isoform α-selective inhibitor under investigation in clinical trials of patients with PIK3CA-altered solid tumors that demonstrated good tolerability (Juric *et al*, 2018). The data presented here show the potent therapeutic effect of PI3Kα inhibition on HO *in vivo*.

# Results

## Inhibition of PI3Kα abrogates SMAD1/5 and mTOR/S6K activation by *Acvr1^{R206H}* in mesenchymal progenitors

Heterotopic endochondral ossification is driven by a resident pool of mesenchymal stem cells at sites of tissue inflammation. It has been shown that fibro/adipogenic precursors (FAPs) are the major cells of origin of heterotopic ossification in both injury-induced and spontaneous HO (Dey *et al*, 2016; Lees-Shepard *et al*, 2018). We generated cultures of mesenchymal stem cells (MSCs) from control mice and infected them with mock virus or with viruses expressing moderate levels of wild-type *Acvr1* (WT), *Acvr1^{Q207D}* (QD), or *Acvr1^{R206H}* (RH) (Fig 1A and B). In agreement with previous data (Hatsell *et al*, 2015), the expression of *Acvr1^{R206H}* altered the signaling properties of ACVR1 in response to activin A as shown by SMAD1/5 phosphorylation (Fig EV1A).

Our group have shown that PI3Kα activity increases cellular levels of SMAD1/5 by reducing its phosphorylation by GSK3 in osteoblasts (Gámez *et al*, 2016). Addition of A66, a PI3Kα-specific inhibitor, reduced SMAD1/5 levels in MSCs, decreased GSK3 phosphorylation, and reduced BMP responsiveness (Fig EV1B). Furthermore, PI3Kα inhibition strongly decreased the expression of osteoblast markers (*Bglap*, *Col1a1*, *Dlx5*, and *Osx/Sp7*) upon osteogenic differentiation of MSCs (Fig EV1C). Therefore, we analyzed the effects of A66 in MSCs transduced with wild-type *Acvr1* or *Acvr1^{R206H}*. Addition of A66 reduced the phosphorylation of GSK3, p38, and S6 independently of the transduced *Acvr1* allele (Fig 1C). Similarly, A66 reduced SMAD1 protein levels and SMAD1/5 phosphorylation. We also analyzed the effects of PI3Kα inhibition on the response to BMP6 and the shifted responsiveness of *Acvr1^{R206H}* upon activin A addition (Hino *et al*, 2015; Alessi Wolken *et al*, 2018). A66 was also able to reduce the levels of pSMAD1/5 upon BMP6 or activin A stimulation in *Acvr1^{R206H}* transduced cells (Fig 1C). These data suggest that PI3Kα inhibition reduces SMAD1/5, AKT, and mTOR/S6K activity in mesenchymal precursors, and impairs responsiveness to canonical BMPs and the acquired response to activin A of *Acvr1^{R206H}*.

## Inhibition of PI3Kα reduces osteochondroprogenitor differentiation potential induced by *Acvr1^{Q207D}* or *Acvr1^{R206H}*

The reduction in the activation of SMAD1/5, p38, and S6K caused by inhibition of PI3Kα suggested that cell progenitors could have decreased skeletogenic potential. To test this hypothesis, we transduced MSCs with wild-type *Acvr1* (WT), *Acvr1^{Q207D}* (QD), or *Acvr1^{R206H}* (RH), and treated them with BMP2, BMP6, or activin A. Expression of the SMAD1/5-dependent gene *Id1* indicated its known sensitivity to either BMP2 or BMP6 (Fig 2, upper panels). More importantly, activin A addition confirms previous observations that the TGFβ/activin A signal through SMAD2/3 inhibits *Id1* expression from endogenous receptors (Lee *et al*, 2005). However, the expression of mutant ACVR1 shifted responsiveness to enhanced *Id1* transcription by activin A. In all conditions, addition of the PI3Kα inhibitor led to a marked decrease in the activation of *Id1* gene expression induced by BMP2, BMP6, or activin A. We also analyzed the expression of chondrocyte and osteoblast markers in these experimental conditions. The PI3Kα inhibitor reduced the expression levels of *Sox9*, a chondrocyte marker, by about 50% and also diminished its induction upon BMP2 addition (Fig 2). Transduction of *Acvr1*-mutant forms also led to increased expression and higher sensitivity to BMP2 and activin A of *Dlx5*, an early osteoblast marker (Acampora *et al*, 1999), *Osx/Sp7*, a osteochondroprogenitor marker (Nakashima *et al*, 2002; Fig 2, lower panels), and increased expression and higher sensitivity to BMP2 of *Bglap*, a osteoblast marker (Fig EV2). Again, the addition of A66 almost abolished these inductive effects of *Acvr1^{R206H}* or *Acvr1^{Q207D}* on *Dlx5*, *Bglap,* and *Osx* expression (Figs 2 and EV2). Altogether, these results demonstrate the ability of PI3Kα inhibitors to reduce the specification of mesenchymal progenitors into skeletal lineages.

## Inhibition of PI3Kα blocks heterotopic ossification in *Acvr1^{Q207D}* mice

To further explore the ability of PI3Kα inhibitors to abrogate HO *in vivo*, we took advantage of transgenic mice expressing the constitutively active allele *ACVR1^{Q207D}* (CAG-Z-EGFP-ACVR1^{Q207D}; Fukuda *et al*, 2006). Since *ACVR1^{Q207Dfl/fl}* mice injected with cardiotoxin and CRE-expressing adenovirus develop robust local HO, this mouse model has been used extensively to search inhibitors of HO (Yu *et al*, 2008; Shimono *et al*, 2011; Agarwal *et al*, 2016). Although A66 potently inhibits PI3Kα *in vitro* (IC$_{50}$ 32 nM), its short half-life in plasma impeded its further clinical development. Structural modifications based on A66 resulted in BYL719 (alpelisib), which possesses optimal PI3Kα selectivity and potency (IC$_{50}$ 4.6 nM), with better pharmacokinetic properties in mice and humans (Fritsch *et al*, 2014; Juric *et al*, 2018; Venot *et al*, 2018). BYL719, which is orally bioavailable, demonstrated a good safety profile in humans, and has been shown to be clinically effective in patients with PIK3CA-related overgrowth syndrome (PROS) and patients with PIK3CA-altered solid tumors (Juric *et al*, 2018; Venot *et al*, 2018). We confirmed that BYL719 also reduced SMAD1/5, AKT, and mTORC1 signaling in MSCs (Fig EV3A). BYL719 was also able to reduce the levels of SMAD1 levels and its phosphorylation upon BMP6 or activin A stimulation in *Acvr1^{R206H}* transduced cells (Fig EV3B) and decrease the expression of osteoblast markers (*Bglap*, *Col1a1*, *Dlx5*, *Osx/Sp7,* and *Runx2*) upon osteogenic differentiation of MSCs (Fig EV3C).

Preclinical oncological studies with PI3Kα inhibitors suggest that intermittent administration might allow higher dosing than continuous treatment, with similar efficacy and lower undesired effects (Fruman *et al*, 2017). Furthermore, preclinical studies in mice

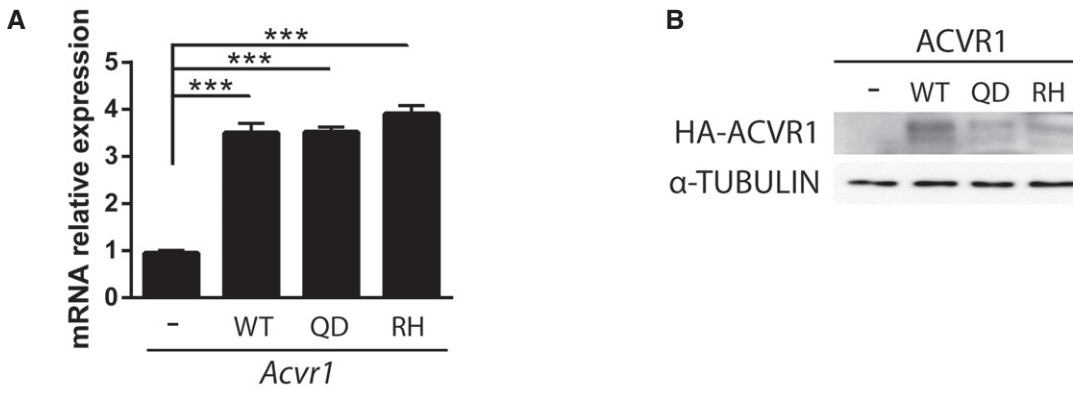

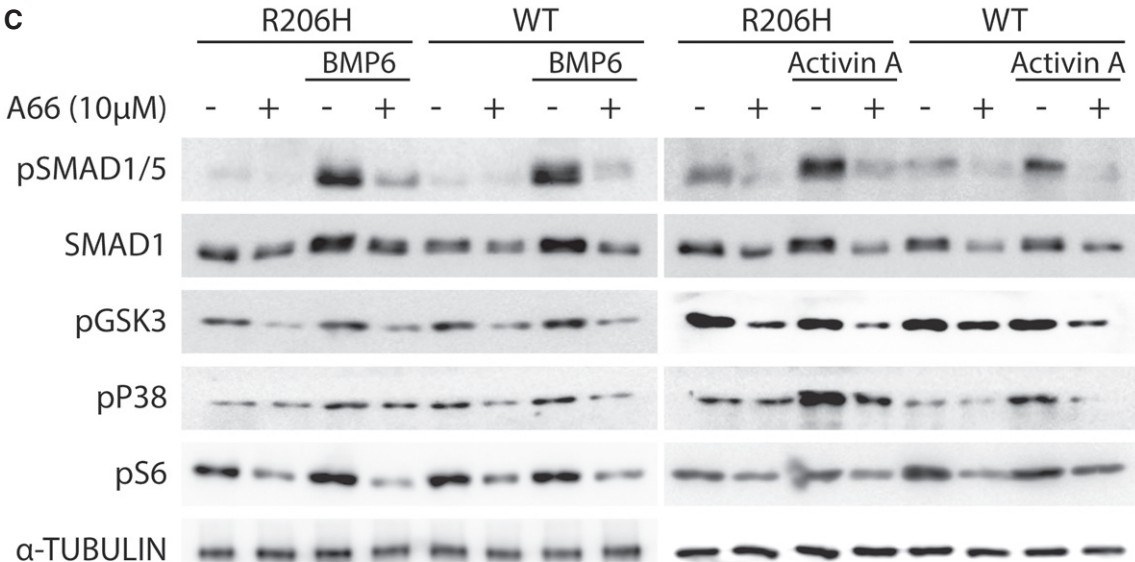

**Figure 1. PI3Kα inhibition reduces SMAD1/5, AKT, and mTOR/S6K activity in MSCs.**

A, B   mRNA and protein levels of MSCs infected with mock virus or with viruses expressing wild-type *Acur1* (WT), *Acur1^Q207D^* (QD), or *Acur1^R206H^* (RH). Data shown as mean ± SEM (*n* = 4 per group). ****P* < 0.001; one-way ANOVA.

C   Immunoblots of MSCs expressing wild-type *Acur1* (WT) or *Acur1^R206H^* (RH) variants. MSCs were serum-starved and treated with 10 μM A66 (PI3Kα inhibitor) for 16 h and then treated with 2 nM BMP6 or 2 nM activin A for 1 h.

Source data are available online for this figure.

indicate that a daily dose of 25–30 mg/kg allows tumor regression and AKT inhibition in tumors, without leading to hyperglycemia and hyperinsulinemia (Fritsch *et al*, 2014). In order to study which regime would be safer and more effective in HO, we investigated two dosing strategies: daily doses of 25 mg/kg of BYL719 and intermittent doses of 25 mg/kg starting 24 h after intramuscular injection of cardiotoxin and CRE adenovirus (Fig 3A). Vehicle-treated *ACVR1^Q207D^* mice developed HO in the injected site by P30 with a high penetrance (13 out of 16 mice; Fig 3B). μCT analysis showed that heterotopic bony tissues were typically embedded in muscle and soft tissues, although apposition to limb skeletal elements was also observed (Fig 3C, Movies EV1–EV3). On the contrary, daily or intermittent administration of BYL719 prevented heterotopic bone formation, and only one animal of each group showed minimal bone formation compared with vehicle-treated mice (Fig 3B–D).

Histological evaluation by H&E, Masson's trichrome, and fast green/safranin O (FGSO) staining revealed that extraskeletal bone lesions contain cartilage, marrow, and bone structures, suggesting an active process of endochondral ossification (Figs 4A and B, and EV4A). Daily or intermittent BYL719-treated mice did not show major bone or cartilage formation in the injected hindlimb at P30 (Figs 4A and B, and EV4). Moreover, H&E staining demonstrated the absence of mesenchymal condensations and fibroproliferative processes, but instead a completed repair of injured skeletal muscle in BYL719-treated *ACVR1^Q207D^* mice (Fig 4B). Altogether, these results suggest that PI3Kα inhibition was effective at suppressing HO in a mouse model, even at their earlier stages.

A drawback of PI3Kα-selective inhibitors is their inevitable, on-target adverse effects on glucose metabolism, as PI3Kα is the main isoform mediating this function (Sopasakis *et al*, 2010; Smith *et al*,

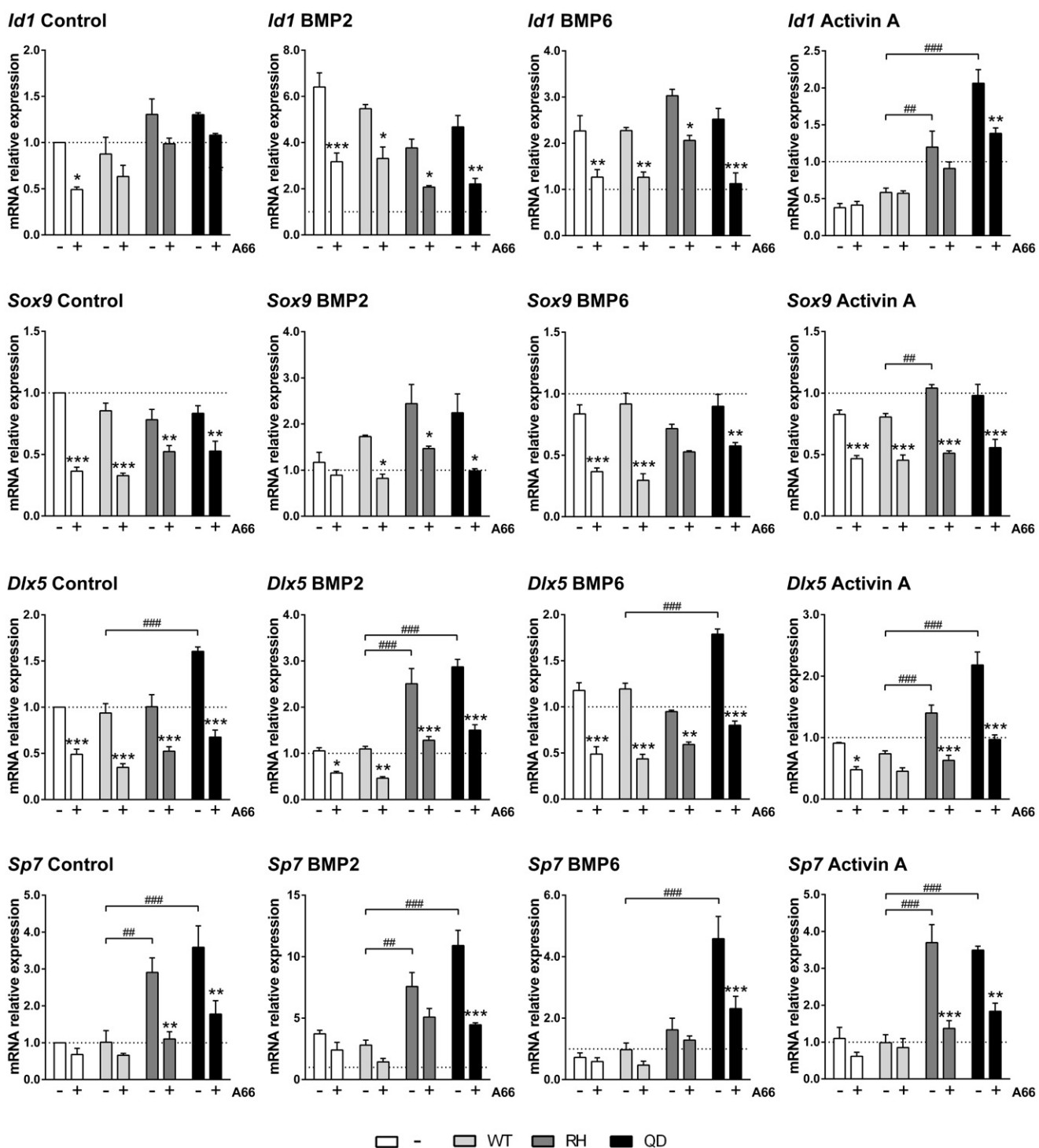

**Figure 2. PI3Kα inhibition reduces the specification of cell progenitors into chondrogenic and osteogenic lineages.**

mRNA expression of osteoblast and chondroblast-specific genes of MSCs expressing wild-type *Acur1*, *Acur1*[R206H], or *Acur1*[Q207D]. MSCs were serum-starved and treated with 10 μM A66 for 16 h and then treated with 2 nM BMP2, 2 nM BMP6, or 2 nM activin A for 2 h. Data shown as mean ± SEM (n = 4 per group). *P < 0.05, ** or ##P < 0.01, *** or ###P < 0.001; two-way ANOVA. Asterisks refer to significance between MSCs treated with or without A66 in each case. Similarly, # refers to significance between *Acur1*[R206H] or *Acur1*[Q207D] MSCs of each group compared to wild-type *Acur1* in cells untreated with A66.

2012). In addition, p110α is the most abundant class IA isoform in bone tissues, and we previously demonstrated its critical role in bone development (Gámez *et al*, 2016). Therefore, we focused on

the effects of BYL719 treatment in metabolic and bone parameters. Although daily dosing of BYL719 led to a significant decrease in body weight of treated animals, intermittent dosage did not decrease

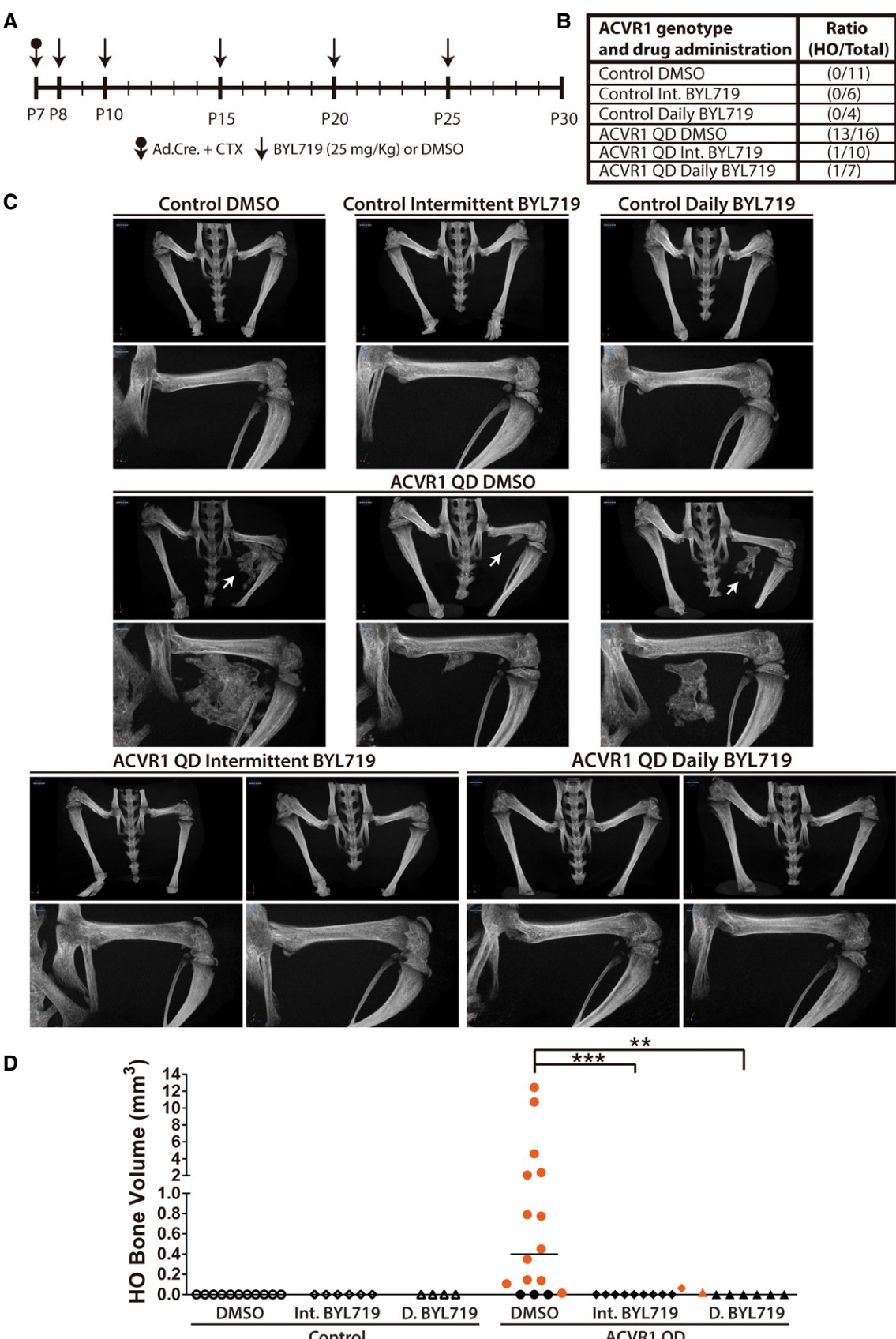

Figure 3.

**Figure 3.   PI3Kα inhibition suppresses heterotopic ossification in mice.**

A   Scheme of the experimental protocol. Heterotopic ossification was induced at P7 with adenovirus-Cre (Ad.Cre) and cardiotoxin (CTX) injection. Control groups only received CTX injection. Drug treatments started on P8. Daily BYL719 (25 mg/kg) or DMSO was administered once a day. Intermittent BYL719 groups received only five (25 mg/kg) BYL719 injections as indicated.

B   Summary of experimental results expressed as ratio between mice with heterotopic ossifications on P30 and the total mice per group. No spontaneous or CTX-induced HO was observed.

C   Representative microtomography frontal 3D images of heterotopic ossification groups. White arrows show heterotopic ossifications. Close-up images of each example show injected left hindlimb.

D   Quantification of heterotopic ossifications of each experimental group. Black symbols indicate the absence of HO. Red symbols indicate HO. Data shown are of each individual mouse with group median. Significance is shown between *Acur1*$^{Q207D}$ groups. **$P < 0.01$, ***$P < 0.001$, Kruskal–Wallis test with Dunn's multiple comparison test.

their body weight or femur length (Fig 5A and B). Similarly, analysis of cortical parameters around the femoral midshaft revealed lower bone perimeter and cortical thickness only in animals treated daily (Fig 5C). These data suggest some effects of daily BYL719 dosage in mice but absent in those mice treated intermittently. Further analysis of femoral trabecular bone parameters gave similar results on bone homeostasis: some osteopenic effects of daily dosing whereas no changes developed in those animals treated intermittently (Fig 5C). Moreover, we found no obvious changes in the circulating levels of glucose or triglycerides, irrespective of the periodicity of the BYL719 treatment (Fig 5D). Therefore, our data suggest that BYL719 has a good therapeutic window for the treatment of heterotopic ossifications, which also agrees with clinical data recently reported in other pathologies (Juric *et al*, 2017, 2018; Venot *et al*, 2018).

## Discussion

HO is a pathological process induced by musculoskeletal trauma or a congenital gain-of-function mutation of the *ACVR1* gene. In both cases, these events converge in an abnormal SMAD1/5 signaling, since inhibition of BMP receptor activity reduces bone formation in HO and FOP mouse models (Yu *et al*, 2008; Agarwal *et al*, 2017). We have identified that PI3Kα inhibitors (A66 or BYL719) block several BMP-activated signaling pathways. Moreover, we have shown that BYL719 treatment rescues the HO phenotype in a mouse model with a promising safety profile. There are several advantages for implementing BYL719 as a therapeutic candidate for HO and FOP: First, inhibition of PI3Kα in mesenchymal progenitors suppresses at least two signaling pathways required for progression of HO downstream of BMP receptors, SMAD1 and mTOR. Mechanistically, we previously showed that PI3Kα inhibitors increased SMAD1/5 degradation by modulation of GSK3 activity, reducing transcriptional responsiveness to BMPs. Accordingly, mice deficient for PI3Kα activity in osteoblasts were osteopenic (Gámez *et al*, 2016). In addition to canonical SMAD1/5, recent reports demonstrated that activation of mTOR, downstream of PI3K/AKT, is also

critical for FOP and non-genetic HO in preclinical models (Hino *et al*, 2017; Qureshi *et al*, 2017). We propose that targeting PI3Kα/AKT has the potential to suppress HO by the inhibition of both pathways.

Whereas *ACVR1*$^{R206H}$ activity is still ligand-dependent (to either BMPs or activins), *ACVR1*$^{Q207D}$ has ligand-independent activity and induces higher chondrogenesis and ossification in functional assays and mouse models of HO (Yu *et al*, 2008; Haupt *et al*, 2014; Dey *et al*, 2016). Our results demonstrated that PI3Kα inhibitors are able to suppress excessive cellular responses induced by classical BMP ligands, as well as the neomorphic signaling by activin A. *ACVR1*$^{Q207D}$ is a well-accepted model of HO (Yu *et al*, 2008; Dey *et al*, 2016), where the mutant allele can be activated post-natally and the phenotype can be rescued by PI3Kα inhibition. Available data indicate some capacity of cells without expression of the mutated allele to contribute to heterotopic lesions (Chakkalakal *et al*, 2012; Dey *et al*, 2016; Lees-Shepard *et al*, 2018). Similarly, non-genetic HO develops in the absence of receptor alterations but with enhanced SMAD and mTOR activities (Agarwal *et al*, 2017; Qureshi *et al*, 2017). We still do not know the relative contribution of the therapeutic effects of BYL719 in normal cells in the lesion environment compared with the effects in mutated cells (expressing the *ACVR1*$^{Q207D}$ allele). In any case, both mutated and wild-type cells within lesions with acquired higher SMAD and mTOR signaling would be targeted by PI3Kα inhibitors (Chakkalakal *et al*, 2012).

PI3Kα inhibitors likely target different cell types required for HO lesion progression. Our data demonstrate that PI3Kα inhibition reduces chondrocyte and osteoblast commitment of MSCs. During cartilage development, PI3K is required for chondrocyte differentiation, survival, and hypertrophy (Beier & Loeser, 2010). The expression of activated AKT in transgenic mice promoted chondrocyte differentiation, whereas a dominant-negative form delayed this process (Rokutanda *et al*, 2009). Furthermore, inactivation of *Pten* in cartilage showed skeletal overgrowth and accelerated chondrocyte hypertrophy (Ford-Hutchinson *et al*, 2007; Ikegami *et al*, 2011). A large body of evidence pointed to PI3Kα function as being also essential for osteoblast specification, bone development, and skeletal maintenance (Liu *et al*, 2007; Guntur & Rosen, 2011;

**Figure 4.   Histological analysis of injected hindlimbs.**

A   Representative images of hematoxylin and eosin (H&E), fast green/safranin O (FGSO), and Masson's trichrome staining in control and *Acur1*$^{Q207D}$ mice. Images were obtained with stereomicroscope (scale bar = 1,000 μm).

B   Close-up representative images of *Acur1*$^{Q207D}$-expressing groups shown in (A). Microscopy images were shown with 2× (scale bar = 1,000 μm) and 4× (scale bar = 500 μm). For FGSO, cartilage is labeled in orange/red. For Masson's trichrome, osteoid is labeled in blue and muscle in red.

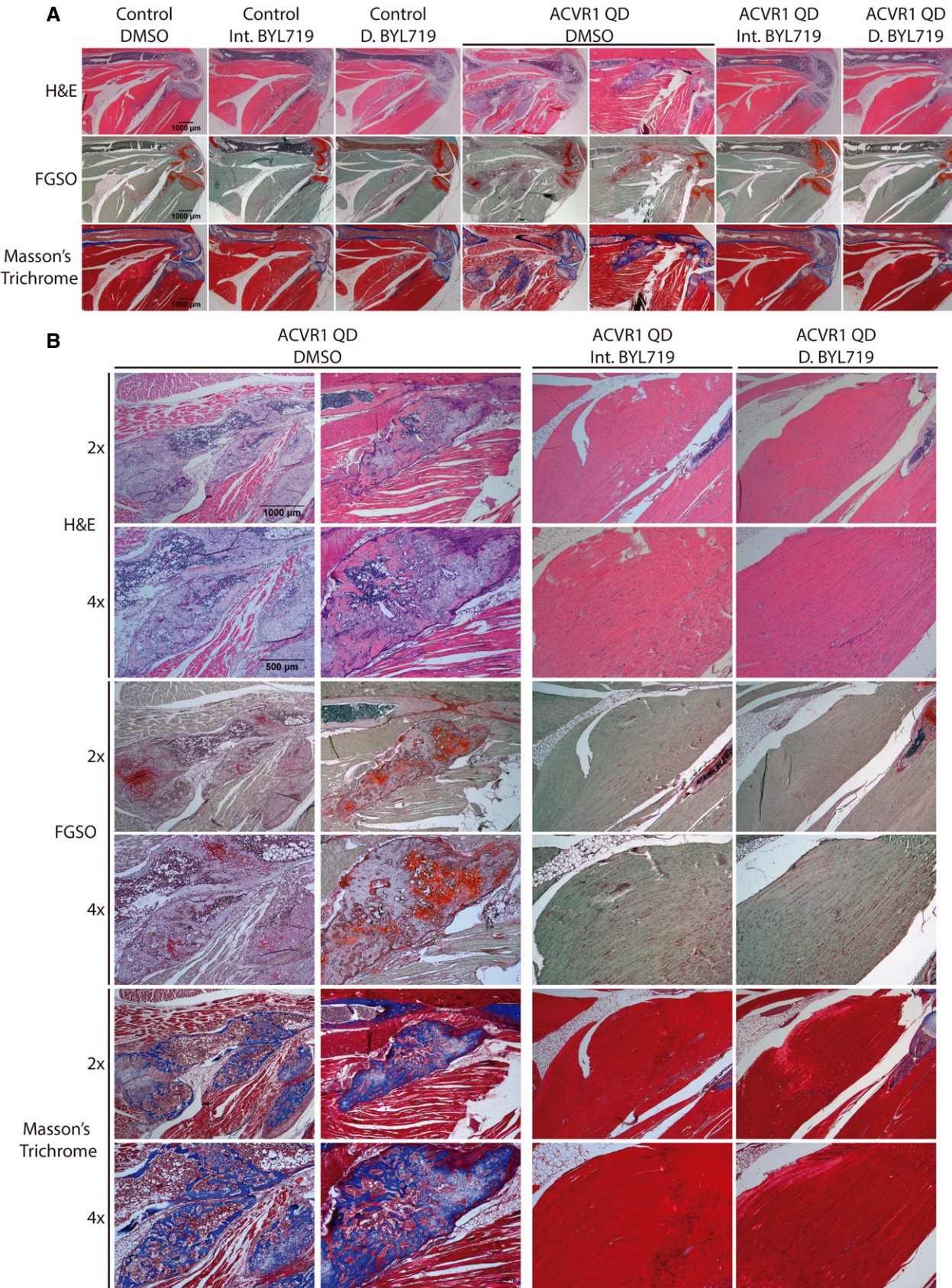

**Figure 4.**

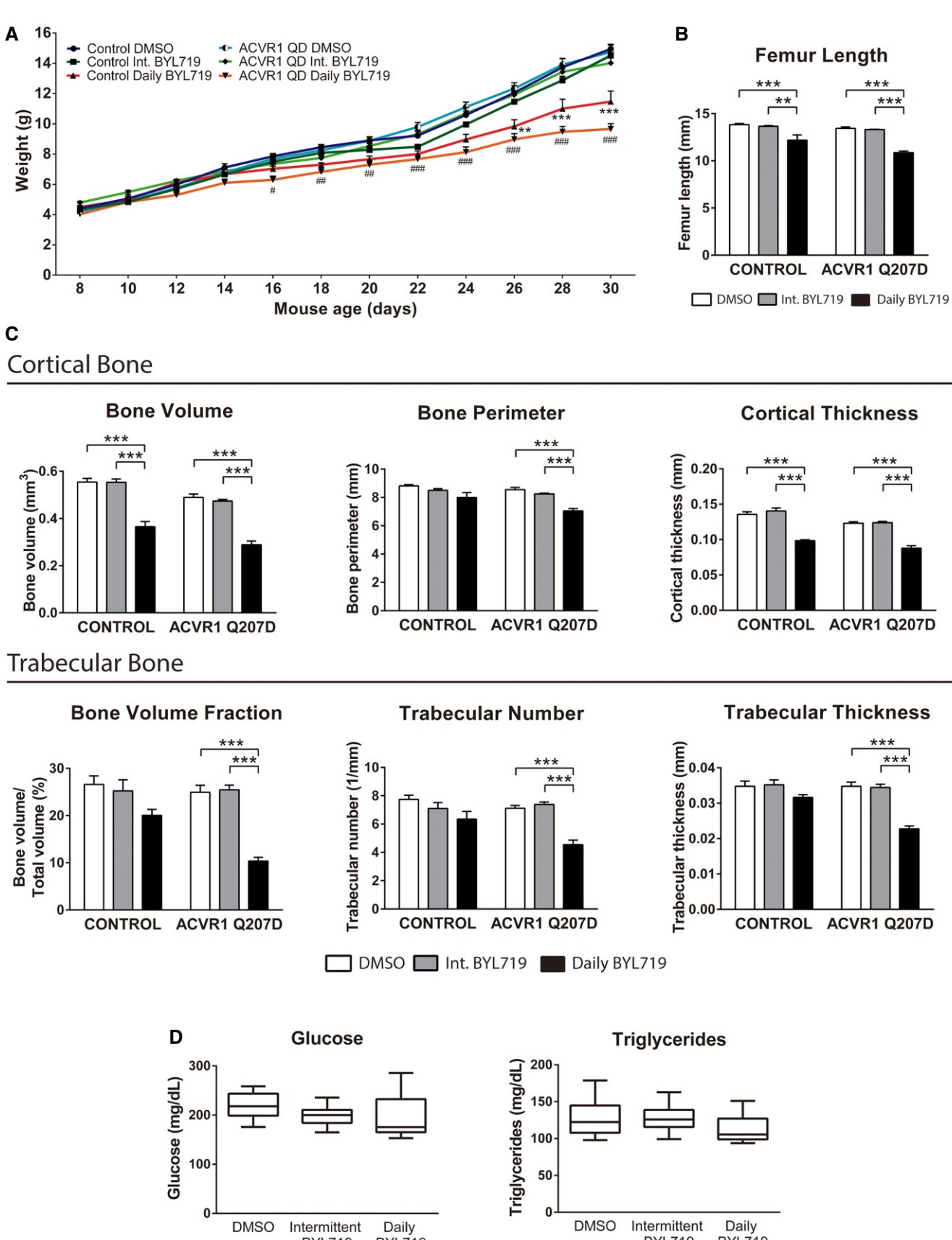

Figure 5.

**Figure 5.  Body weight, cortical bone, trabecular bone, and glucose/triglycerides of BYL719 treated mouse.**
A    Body weight of mice from P8 (start of treatment with DMSO or BYL719) to P30 (experimental endpoint). Data shown as mean $\pm$ SEM (*n* per group detailed in Fig 3). $^{\#}P < 0.05$, ** or $^{\#\#}P < 0.01$, *** or $^{\#\#\#}P < 0.001$; two-way ANOVA. Asterisks refer to significance between *Acur1*$^{Q207D}$ groups (treated daily with BYL719) compared to Acvr1$^{Q207D}$ treated with DMSO. Similarly, # refers to significance between groups treated daily with BYL719, compared to control group treated with DMSO.
B, C  Femur length, cortical bone, and trabecular bone parameters of different mice groups at P30. Asterisks refer to significance between *Acur1*$^{Q207D}$ groups (treated with daily or intermittent BYL719) compared to Acvr1$^{Q207D}$ treated with DMSO and between control groups (treated with daily or intermittent BYL719) compared to control group treated with DMSO. Data shown as mean $\pm$ SEM (*n* per group detailed in Fig 3). **$P < 0.01$, ***$P < 0.001$; two-way ANOVA.
D    Glucose and triglyceride levels at P30 of mice fed *ad libitum*. Data shown as box and whisker plot. Median is plotted, and box extends from 25th to 75th percentiles. Whiskers go from Min to Max. One-way ANOVA.

Xian *et al*, 2012; Gámez *et al*, 2016). Similarly, angiogenesis selectively requires PI3Kα to control endothelial cell migration (Graupera *et al*, 2008). Tissue inflammation has been strongly associated with the onset of HO, and depletion of mast cells and macrophages has been shown to impair HO in FOP mouse models (Shore & Kaplan, 2010; Convente *et al*, 2018). Since PI3Kγ and PI3Kδ are the dominant isoforms and the role of PI3Kα is extremely restricted in immune cells, it is less likely a direct effect of BYL719 on macrophage or mast cell function (Fruman *et al*, 2017). However, we cannot exclude an indirect effect in the recruitment of inflammatory cells and the establishment of a permissive tissue microenvironment for HO progression. Finally, BYL719 demonstrated a good tolerability and safety profile in humans and has been shown to be clinically effective in patients with PIK3CA-related overgrowth syndrome (PROS) and patients with PIK3CA-altered solid tumors (Juric *et al*, 2018; Venot *et al*, 2018). This evidence resulted in its recent approval by the FDA for the treatment of PIK3CA-mutant advanced or metastatic breast cancer. Therefore, BYL719 has been well evaluated regarding its pharmacokinetics, administration, tolerated dose, and side effects in humans and mice (Fritsch *et al*, 2014; Juric *et al*, 2018; Venot *et al*, 2018). Daily oral doses of 400 mg of BYL719 were well tolerated by patients in a phase Ia study in PIK3CA-altered solid tumors (Juric *et al*, 2018), whereas PROS patients, after 18 months, are still been treated daily with 250 mg of BYL719 without major complications (Venot *et al*, 2018). The recommended BYL719 dose approved by the FDA is 300 mg (two 150 mg film-coated tablets) taken orally, once daily. In these studies, the most frequent adverse effect was peripheral insulin resistance and hyperglycemia, an effect of PI3Kα inhibition that could be managed by concomitant oral antidiabetic medication (Juric *et al*, 2018; Venot *et al*, 2018). In mice, pharmacokinetic studies showed that a daily dose of BYL719 of 25 mg/kg is sufficient to keep more than 90% of AKT activity inhibited and to abrogate growth of grafted tumors (Fritsch *et al*, 2014). These studies also demonstrate a window of therapeutic opportunity without adverse effects between 10 and 40 mg/kg (Fritsch *et al*, 2014). Thus, repurposing BYL719 for the treatment of HO, at least during a restricted temporal window (i.e., soft tissue trauma and surgery to remove ectopic bone in HO, or during flare-ups in FOP patients), should pose few contraindications. Our work in young mice (from P7 to P30) showed that mice treated daily with 25 mg/kg of BYL719 had normoglycemia but have reduced growth rate and developed osteopenia over time. However, those treated with intermittent dosage did not show any detectable adverse effect while maintaining therapeutic efficacy. Intermittent dosing has been yielding encouraging data for PI3K inhibitor therapeutic strategies in humans (Fruman *et al*, 2017). For instance, intermittent dosing of PI3Kα inhibitors in clinical trials allowed more pronounced acute

PI3K/mTOR pathway inhibition with fewer adverse effects in normal tissues (Ma *et al*, 2016; Juric *et al*, 2017). Over the last years, beyond anti-inflammatory drugs, potential treatments for FOP have been developed and are currently in ongoing clinical trials. Some treatments focus on ACVR1 activation (e.g., anti-activin A antibody or ACVR1 kinase inhibitors), while others, such as the PI3Kα inhibitors presented here, target cells and signaling pathways required for heterotopic ossification downstream ACVR1 (e.g., palovarotene or rapamycin; Wentworth *et al*, 2019). *ACVR1* is almost ubiquitously expressed, and the pathways targeted by these drugs are also involved in multiple physiological and developmental processes. Therefore, in addition to therapeutic efficacy, side effects in multiple tissues must be carefully monitored. Noteworthy, BYL719 has demonstrated a good safety profile in children suffering PROS treated daily for more than a year (Venot *et al*, 2018). However, it remains to be investigated whether BYL719 can have any undesirable effect specifically related to the abnormal SMAD1 signaling downstream *ACVR1*$^{R206H}$ in FOP patients. In addition, further studies should be performed to optimize dosing and timing of administration or to evaluate combinatorial approaches with other pharmacological agents (i.e., anti-inflammatories).

## Materials and Methods

### Murine bone marrow mesenchymal stem cell isolation and culture

Murine bone marrow-derived mesenchymal stem cells (BM-MSCs) were isolated from 6- to 8-week-old C57BL6/J mice as previously described (Soleimani & Nadri, 2009). Mice were euthanized, and femurs were dissected and stored in DMEM (Biological Industries) with 100 U/ml penicillin/streptomycin (P/S). Under sterile conditions, soft tissues were cleaned and femur ends were cut. Bone marrow was strongly flushed with media using a 27-gauge needle, and cell suspension was filtered through a 70-μm cell strainer (BD Falcon) and seeded in a 100-mm cell culture dish. Non-adherent cells were discarded after 3 h. Media were slowly replaced every 12 h for up to 72 h. Then, media were replaced every 2 days until 70% confluence was reached. At this stage, cells were washed with warm PBS and lifted by incubation with 0.25% trypsin/0.02% EDTA for 5 min at room temperature. Lifted cells were cultured and expanded.

### Cell culture

Murine BM-MSCs were cultured in DMEM supplemented with 10% FBS, 2 mM L-glutamine, 1 mM sodium pyruvate, and 100 U/ml P/S

(Biological Industries) and incubated at 37°C with 5% $CO_2$. For osteogenic differentiation, cells were cultured in α-MEM with 10% FBS, 2 mM ʟ-glutamine, 1 mM sodium pyruvate, 50 μM ascorbic acid, 10 mM β-glycerophosphate, and 100 U/ml P/S (Biological Industries) for 14 days.

Unless otherwise stated, 10 μM A66 (Selleckchem) or 10 μM BYL719 (Chemietek) was used for 16 h, in the absence of FBS, prior to cell harvest. 2 nM BMP2, 2 nM BMP6, and 2 nM activin A (R&D) were used for 2 h (qRT–PCRs) or 1 h (Western blot).

### Retroviral transduction

Bone marrow mesenchymal stem cells from control mice were infected with mock virus (pMSCV-GFP) or with viruses expressing wild-type *Acvr1* (WT), *Acvr1*[Q207D] (QD), or *Acvr1*[R206H] (RH). Plasmids with WT, QD, and RH *Acvr1* forms were kindly provided by Dr. Petra Seemann and were subcloned into pMSCV viral vector. *Acvr1* expression levels were analyzed by qRT–PCR and Western blot.

### Western blot

Cell lysates were resolved on PAGE and transferred to Immobilon-P membranes (Millipore). Primary antibodies against SMAD1 (Cell Signaling, 6944), p-SMAD1/5 (Cell Signaling, 9516), p-GSK3 (Cell Signaling, 9331), p-P38 (Cell Signaling, 9211), p-S6 (Cell Signaling, 2211), HA (Roche, 11666606001), and α-TUBULIN (Calbiochem, CP06) were used at 1:1,000 dilution. Antibody against β-ACTIN (Abcam, Ab6276) was used at 1:4,000 dilution. Antibody against ID1 (Santa Cruz Biotechnology, C-20) was used at 1:200 dilution. Binding was detected with horseradish peroxidase-conjugated secondary antibodies (Sigma). Immunoblot images were obtained with Fujifilm LAS3000.

### Gene expression analysis

Total RNA was extracted from BM-MSCs using TRIsure reagent (Bioline). At least 2 μg of purified RNA was reverse-transcribed using the High-Capacity cDNA Reverse Transcription Kit (Applied Biosystems). Quantitative PCRs were carried out on ABI Prism 7900 HT Fast Real-Time PCR System with TaqMan 5′-nuclease probe method (Applied Biosystems) and SensiFAST Probe Hi-ROX Mix (Bioline). All transcripts were normalized using *Tbp* as an endogenous control (Appendix Table S1).

### Heterotopic ossification mouse model

To study heterotopic ossification *in vivo*, we used Cre-inducible constitutively active ACVR1[Q207D] (CAG-Z-EGFP-caALK2) mouse model as previously described (Fukuda *et al*, 2006; Yu *et al*, 2008; Shimono *et al*, 2011). We analyzed mice of both genders with a C57BL/6 background. In order to induce heterotopic ossification in P7 ACVR1[Q207D] mice, $1 \times 10^8$ pfu of adenovirus-Cre (Ad-CMV-Cre, Viral Vector Production Unit, UAB) and 0.3 μg of cardiotoxin (L8102, Latoxan) in 10 μl 0.9% NaCl volume were injected into the left hindlimb. Control groups had the same procedure without Ad-Cre in the injection. On P8, mice started receiving either placebo

(intraperitoneal administration (i.p.) of DMSO) or BYL719 (i.p. of 25 mg/kg), both diluted in 0.5% carboxymethylcellulose sodium (Sigma). Treatment groups received BYL719 either daily or with intermittent administration (see Fig 4A). Mice were housed under controlled conditions (12-h light/12-h dark cycle, 21°C, 55% humidity) and fed *ad libitum* with water and a 14% protein diet (2014 Teklad, Envigo). Mice were regularly weighed over the whole period. For *in vivo* studies, experimental endpoint was determined to be at P30, when most of the untreated mice had presented heterotopic ossification. Primers used to detect mice genotype were 5′-GTGCTGGTTATTGTGCTGTCTC-3′ and 5′-GACGACAGTATCGGCCTCAGGAA-3′. All procedures were approved by the Ethics Committee for Animal Experimentation of the Generalitat de Catalunya.

### Micro-computed tomography analysis

Caudal half of mice were collected and fixed in 4% paraformaldehyde (PFA) for 48 h at 4°C. Specimens were conserved in PBS and high-resolution images were acquired using a computerized microtomography imaging system (SkyScan 1076, Bruker microCT), in accordance with the recommendations of the American Society of Bone and Mineral Research (ASBMR). Samples were scanned in air at 50 kV and 200 μA with an exposure time of 800 ms, using a 1-mm aluminum filter and an isotropic voxel size of 9 μm. Two-dimensional images were acquired every 1° for 180° rotation and subsequently reconstructed, analyzed for bone parameters, and visualized by NRecon v1.6, CT-Analyser v1.13, and CTVox v3.3 programs (Bruker), respectively. For heterotopic ossification, manual VOIs comprising heterotopic ossifications were employed and a binary threshold was established at 25–255. For trabecular and cortical evaluation, non-injected hindlimbs were collected and femurs were analyzed with the parameters detailed above. For trabecular measurements, manual VOIs starting 100 slices from distal growth plate of the femur and extending to the diaphysis for 150 slices were employed and a binary threshold was established at 20–255. For cortical measurements, manual VOIs of 100 slices delineating the femur medial cortex around the femoral midshaft were employed and a binary threshold was established at 50–255.

### Histology

Whole legs were fixed in 4% PFA for 48 h at 4°C, decalcified in 16% EDTA pH 7.4 for 6 weeks, and embedded in paraffin. 5-μm sections were cut and stained with hematoxylin and eosin, fast green/safranin O, or Masson's trichrome (PanReac AppliChem). For whole leg visualization, images were obtained with Brightfield SteREO Lumar v.12 stereomicroscope (Carl Zeiss). For higher magnifications, images were obtained with Brightfield Eclipse E800 (Nikon).

### Blood tests

Blood samples were collected by cardiac puncture at sacrifice, and centrifuged at $1,850 \times g$ for 15 min at 4°C to obtain serum samples. Aliquots were analyzed at Clinical Biochemistry Service of the Faculty of Veterinary Medicine (Universitat Autònoma de Barcelona).

**The paper explained**

**Problem**

Heterotopic ossification (HO) is characterized by ectopic bone formation in soft tissues at extraskeletal sites. Trauma-induced heterotopic ossification develops as a common post-operative complication or after mechanical trauma. Clinical therapy is now limited to anti-inflammatory drugs, radiation, or surgical excision of the already formed bone. In addition, congenital heterotopic ossification could arise from fibrodysplasia ossificans progressiva (FOP). FOP patients, which present a mutant form of the BMP receptor ACVR1, have episodic flare-ups usually associated with inflammation that end up forming ectopic bone. Cumulative effects of these osteogenic events include progressive immobility and a shorter life span. Clinical therapy is now limited to symptomatic treatments. Therefore, the development of new therapies for heterotopic ossification treatment is required.

**Results**

We studied the effect of PI3Kα inhibition in osteogenesis of murine mesenchymal precursors *in vitro*. PI3Kα inhibition impairs responsiveness to canonical bone morphogenetic proteins and the acquired activin A response of mutant ACVR1, reducing the specification of cell progenitors into skeletal lineages. We also studied the effect of BYL719, a PI3Kα inhibitor, in a genetic mouse model of heterotopic ossification with two different dosages: daily and intermittent. Mice treated daily or intermittently with BYL719 did not show ectopic bone or cartilage formation. Furthermore, the intermittent treatment with BYL719 was not associated with any substantial side effect.

**Impact**

This work provides evidence that PI3Kα inhibition is beneficial for heterotopic ossification, a pathology without therapeutic treatment. Our data show that BYL719 can prevent ectopic bone or cartilage formation in mouse. In addition, BYL719 has already demonstrated a good tolerability and safety profile in humans. Therefore, although further studies should be performed to optimize dosing and timing of administration or to evaluate combinatorial approaches with other pharmacological agents, our work suggests that PI3Kα inhibition is a potential therapeutic strategy for heterotopic ossification.

## Statistical analysis

The results were expressed as mean ± SEM. Unless stated otherwise in each figure legend, statistical analysis was performed using ANOVA. Detailed *P*-values are provided in Appendix Table S2. Unless stated otherwise in each figure legend, results were replicated four times with technical triplicates. Heterotopic ossification induction was performed blinded to mouse and group identity. The evaluation of heterotopic ossification formation and quantification was performed independently and blinded to mouse and treatment group. Statistical tests were performed on GraphPad Prism 6 (GraphPad Software Inc.).

**Expanded View** for this article is available online.

## Acknowledgements

We thank E. Adanero, E. Castaño, L. Gómez-Segura, L. Mulero, and B. Torrejón for technical assistance and G. Sanchez-Duffhues for critical reading of the article. C. Sánchez-de-Diego is the recipient of a F.P.U. fellowship from the Spanish Ministry of Education. This work was supported by grants from the MEC (BFU 2014-56313 and BFU201782421-P) to F. Ventura.

## Author contributions

JAV, BG, JLR, and FV designed the study, and analyzed and interpreted the results. JAV, CS-D, and BG conducted the experiments. YM provided the *Acur1^{Q207D}* mice and interpreted the results. JAV and FV wrote the article. All authors contributed to the article. FV supervised the project.

## Conflict of interest

The authors declare that they have no conflict of interest.

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
