## [Review Process File · EMBO Molecular Medicine]

Inhibition of phosphatidylinositol 3-kinase α (PI3K α) prevents heterotopic ossification

José Antonio Valer, Cristina Sánchez-de-Diego, Beatriz Gámez, Yuji Mishina, José Luis Rosa and Francesc Ventura

Review timeline:	Submission date:	6 March 2019
	Editorial Decision:	8 April 2019
	Revision received:	4 June 2019
	Editorial Decision:	1 July 2019
	Revision received:	8 July 2019
	Accepted:	10 July 2019

Editor: Céline Carret

Transaction Report:

1st Editorial Decision

8 April 2019

Thank you for the submission of your manuscript to EMBO Molecular Medicine. We have now heard back from two out of the three referees whom we asked to evaluate your manuscript. As their reports are supportive and overlapping, we cannot justify waiting any longer for the last referee to provide her/his report.

You will see that both referees find the study to be of interest and recommend further consideration should you address their set of comments. More details should be provided, along with explanations and better discussions. The quality of the western blots should be improved, and the therapeutic implications made clearer. I would also like to encourage following referee 2 suggestion for markers and the experiment should be repeated accordingly.

We would therefore welcome the submission of a revised version within three months for further consideration and would like to encourage you to address all the criticisms raised as suggested to improve conclusiveness and clarity. Please note that EMBO Molecular Medicine strongly supports a single round of revision and that, as acceptance or rejection of the manuscript will depend on another round of review, your responses should be as complete as possible.

I look forward to receiving your revised manuscript.

***** Reviewer's comments *****

Referee #2 (Comments on Novelty/Model System for Author):

Technical: The Western blots are of low quality as proteins that should be present in control samples are hardly visible, making changes that occur in the treatment groups hard to interpret.

Novelty: Others have examined the role of PI3K inhibition on ectopic bone formation.

Impact: As there are no current approved therapies for FOP, potential for impact is high.

Referee #2 (Remarks for Author):

This MS presents interesting findings about the potential for PI3 kinase inhibitors as a treatment for FOP and other forms of HO. However, as presented several questions arise upon review:

1. Why is intermittent treatment free of any substantial side effects that would occur with reduced BMP signaling, particularly in lung and kidney? Since the authors provide no rationale for the intermittent strategy, there is no way to know that efficacy will translate from mouse to humans.
2. Several of the Western blots show no/absent Smad1 protein in control lanes... this suggests technical issues with the blots as Wt cells should have Smad 1. It is not possible to conclude that signaling is changed if there is no Smad 1 in the control lane.
3. Osx/Sp7 is described as an osteoblast marker when it is actually an osteoprogenitor marker. That being the case, the treatment seems to be effective at blocking the initial steps in HO and then would only be useful before a flare-up as a prophylactic. In that case, there is no rescue from HO but prevention of HO.

Referee #3 (Remarks for Author):

In this paper the authors test the hypothesis that inhibition of PI3K α signaling could be used to treat heterotopic ossification. The results of this proof of principle experiment are convincing in a mouse model of heterotopic ossification and are not over-interpreted. There are few concerns that the authors should address .

- The choice of osteoblastic markers is unusual. Col1a1, Bsp, Osteocalcin, Runx2 and Sp7 should be used for all experiments. As for chondrocytes they should use Col2a1, Aggrecan and Sox9.

- Some of the effects of A66 on Smad phosphorylation are modest or non existent (Figure1C, supplemental Figure1); given these modest effects the authors should consider other hypotheses in discussing the mode of action of inhibiting PI3K α pathway.

-All experiments with A66 are performed with 10microMolar except the one in supplemental figure1c, what is the reason for that?

1st Revision - authors' response

4 June 2019

Reviewer # 2 Specific points:

Point #1. "Why is intermittent treatment free of any substantial side effects that would occur with reduced BMP signaling, particularly in lung and kidney? Since the authors provide no rationale for the intermittent strategy, there is no way to know that efficacy will translate from mouse to humans."

Most clinical studies employ daily administration strategies, with drug doses adjusted to tolerated doses. Although this approach ensures a reasonable safety profile, they have the risk of not reaching an effective dose in the target cells. Alternatively, high-dose intermittent therapy, while sparing normal tissues, has been shown to be better for PI3K/mTOR inhibitors in preclinical and clinical applications in oncology by leading to a more complete pathway inhibition (Fruman et al., 2017: PMID:28802037). The clinical success of this strategy relies on the ability of normal tissues to recover during inter-dose intervals. For instance, PI3K α inhibitors were more effective when administered intermittently in preclinical breast cancer models (Liu et al., 2013: PMID:24170767; Hudson et al., 2016: PMID:26839307). Intermittent dosing of HER2 or PI3K α inhibitors also showed more pronounced pathway inhibition with equivalent toxicity in clinical oncology trials

(Juric *et al.*, 2017: PMID:28490463; Morikawa *et al.*, 2019: PMID:30988080). This seems to be also the case of BYL719 in our HO murine model. The same dose of BYL719, when administered intermittently, had the same therapeutic efficacy for HO but had no side effects on bone tissue or body weight (Figure 5). However, additional studies should be performed to analyze possible side effects in other tissues. Discussion of the rationale behind the intermittent strategy is included in the discussion.

Point #2. “Several of the Western blots show no/absent Smad1 protein in control lanes... this suggests technical issues with the blots as Wt cells should have Smad 1. It is not possible to conclude that signaling is changed if there is no Smad 1 in the control lane.”

We improved the quality of the western blots and repeated some experiments. New western blots are included in the Figures 1C and the whole Figure EV1B has been repeated. Moreover, we also analyzed the effects of BYL719 on SMAD1 levels and its phosphorylation upon BMP6 or Activin A addition (Figure EV3B). Altogether, the data confirms that PI3Ka inhibitors reduce SMAD1 levels and its phosphorylation upon ligand activation.

Point #3. “Osx/Sp7 is described as an osteoblast marker when it is actually an osteoprogenitor marker. That being the case, the treatment seems to be effective at blocking the initial steps in HO and then would only be useful before a flare-up as a prophylactic. In that case, there is no rescue from HO but prevention of HO.”

*Osx/Sp7 function is required for bone development and homeostasis where it has a major role in osteochondroprogenitor specification, osteoblast gene expression, and in osteocyte maturation and function (Zhou *et al.*, 2010; PMID:20615976). To analyze further the ability of PI3Ka inhibitors to block osteoblast differentiation and function, we include data regarding expression of additional osteoblast-specific genes (Col1a1, Bglap, and Runx2) upon Acvr1 mutant expression in the new Figure EV2 in addition to those previously reported in Figure 2. Moreover, in addition to Dlx5, Runx2, Sp7, and Col1a1, expression of additional markers (Bglap and Id1) is also included in the analysis of the effects of PI3Ka inhibitors on osteoblast differentiation of BM-MSCs (new panels in the Figure EV1C).*

Reviewer # 3 Specific points:

Point #1. “The choice of osteoblastic markers is unusual. Col1a1, Bsp, Osteocalcin, Runx2 and Sp7 should be used for all experiments. As for chondrocytes they should use Col2a1, Aggrecan and Sox9.”

We analyzed and included new data regarding expression of additional specific marker genes (Col1a1, Bglap (Osteocalcin), Runx2, and Col2a1) upon Acvr1 mutant expression in the new Figure EV2 in addition to those previously reported in Figure 2 (Id1, Sox9, Dlx5, and Sp7). Moreover, expression of the additional markers (Bglap and Id) is also included in the analysis of the effects of PI3Ka inhibitors on osteoblast differentiation of BM-MSCs (new panels in the Figure EV1C). Expression of the late chondrocyte and late osteoblast genes Aggrecan and Ibsp remained below detection limits in our model of BM-MSC cultures.

Point #2. “Some of the effects of A66 on Smad phosphorylation are modest or non existent (Figure1C, supplemental Figure1); given these modest effects the authors should consider other hypotheses in discussing the mode of action of inhibiting PIK3alpha pathway.”

We improved the quality of the western blots and repeated some experiments. New western blots are included in the Figures 1C and the whole Figure EV1B has been repeated. Moreover, we also analyzed the effects of BYL719 on SMAD1 levels and its phosphorylation upon BMP6 or Activin A addition (Figure EV3B). Altogether, the data confirms that PI3Ka inhibitors reduce SMAD1 levels and its phosphorylation upon ligand activation. In addition, PI3Ka inhibitors also reduce pGSK3 and pS6 levels, suggesting inhibition of AKT and mTOR/S6K signaling. In the discussion section, we propose that targeting PI3Ka have the potential to suppress HO by inhibition of these pathways (SMAD1/5 and mTOR). However, we cannot exclude modulation of additional signaling pathways relevant for HO.

Point #3. “All experiments with A66 are performed with 10microMolar except the one in supplemental figure1c, what is the reason for that?”

In most experiments, analysis of the effects of PI3Ka inhibitors on gene expression and signaling pathway activation were analyzed upon ligand activation (BMP2, BMP6, or Activin A) for

2h (qPCR) and 1h (western blot) in cells pretreated for 16h with A66 or BYL719. In the experiment shown in supplemental Figure 1C, MSCs were induced to differentiate during 14 days in the presence of 50 μ M ascorbic acid and 10 mM β -glycerophosphate. In this experiment, treatment with A66 was constant in the media throughout the whole period (14 days), therefore, we reduced the dose of A66 to avoid undesired possible toxic effects.

Reviewer # 1 Specific points:

Point #1. “There is a major disconnect between the use of A66 as the inhibitor with which the majority of the in vitro work was carried out, and the inhibitor used for the in vivo experiments, BYL719. If it is the latter to be used as the drug, shouldn't the focus of the in vitro work be on that one? Furthermore, from the in vitro experiments that are shown on Figure EV2, there appear to be differences on the actions of the two drugs. For example, in cells treated with BMP2, the levels of Smad1 protein appear to increase, whereas this is not the case with A66. Moreover, there is barely a discernible effect of BYL719 on BMP2-induced Smad1/5 phosphorylation, and pGSK3 levels are also barely affected. Is this variation from experiment to experiment or is it a real effect? The authors need to go back and repeat the key experiments using BYL719, especially if they want to make a case about mechanism of action. Alternatively, the paper could be rewritten with a focus on the pharmacologic data (using prior findings with genetic ablation of p110a as the starting point), and the mechanistic aspects can be explored later or provided as supplementary data, or with the corresponding claims toned down.”

We improved the quality of the western blots and repeated some experiments (Figure EV1B). New western blots are included in the Figures 1C and EV1B. The results confirm that addition of BMP2 increase the levels of SMAD1 as it was previously described (Gámez et al., 2016 PMID:26896753). More importantly, we performed new assays to clarify the mechanisms of action of BYL719. We analyzed the effects of BYL719 on SMAD1 levels and its phosphorylation upon BMP6 or Activin A addition (new Figure EV3B). In addition, we analyzed by qPCR the expression of bone markers after infection of BM-MSCs with different mutant Acvr1 receptors and BYL719 addition (new Figure EV3C). The data suggests the same mechanisms of action for A66 and BYL719.

Point #2. “The authors conflate mouse models of HO with mouse models of FOP; although one can argue that the final outcome is the same (i.e. heterotopic bone generated via an endochondral process), the mechanism that drives each one is different. In trauma induced HO (also commonly referred to as “non-genetic”), osteogenic BMPs and their receptors appear to be the main drivers. In FOP, HO results from activation of FOP mutant ACVR1 by Activin A, a ligand that in wild type animals is neither osteogenic nor does it activate ACVR1. The Q207D model utilized here is not a bona fide mouse model of FOP – it is a model of HO. The reason that it induces HO is because expression of ACVR1[Q207D] results in activation of Smad1/5/8 signaling. Although the receptor being utilized is ACVR1 (which is also the gene mutated in FOP), the same effect would be observed if one were to activate Smad1/5/8 signaling by any other means. The ACVR1[Q207D] mouse model has been superseded by two other models (PMID 26333933 and 29396429) that genetically model none other than the R206H variant of ACVR1 (which is the variant found in the overwhelming majority of FOP patients that have been genetically ascertained to date). Therefore, claims that ACVR1[Q207D] is a mouse model of FOP should be toned down if not removed entirely. It should be clear that what has been studied here is not FOP but rather a laboratory version of inducing HO by an endochondral process.”

As suggested by the reviewer, we avoid referring the ACVR1Q207D mice as a model of FOP throughout the manuscript. Instead, we refer to them as an inducible model of heterotopic ossification.

Point #3. “The Discussion should include a section comparing the different potential treatments for non-genetic HO as well as those being developed for FOP (currently three of them are undergoing clinical trials – Palovarotene; anti-Activin A monoclonal antibody; Rapamycin – and at least one more, an ALK2 kinase inhibitor is about to enter the clinical development stage) with the strategy being proposed here.”

We included a section in the last part of the discussion introducing the distinct potential treatments that are in ongoing clinical trials. All the new therapies, including BYL719, target the activity of ACVR1 or signaling pathways that are almost ubiquitously expressed. Therefore, we emphasized that, in addition to their therapeutic efficacy, side effects in multiple tissues and organs must be carefully monitored.

Point #4. “In the Discussion section, the authors tout the safety of BYL719 based on clinical experience thus far. However, a cautionary note is in order in that treatment of FOP with this drug may have FOP-specific side effects. This possibility should not be discounted.”

We included a sentence at the end of the discussion to introduce this possibility. We included the sentence “However, it remains to be investigated whether BYL719 can have any undesirable effect specifically related to the abnormal SMAD1 signaling downstream ACVR1R206H in FOP patients”.

Minor comments:

Point #1. “In the Introduction, the authors state that “Enhanced BMP signaling in patients and mouse models of FOP has been attributed to loss of auto-inhibition of the receptor and increased responsiveness to BMP ligands (Groppe et al, 2007; Van Dinther et al, 2010; Chaikuad et al, 2012). In addition, recent reports demonstrated that ACVR1 mutations in FOP are neomorphic, abnormally transducing BMP signals in response to activin A (Hatsell et al, 2015; Hino et al, 2015).” This is a misleading interpretation of the cited literature on multiple counts:

a. There is no evidence that it is “enhanced BMP signaling in patients” that drives the pathology of FOP – the evidence is circumstantial and based on in vitro findings.

b. The same applies to mouse models of FOP. These models clearly show that osteogenic BMPs play little if any role in the main pathology of FOP, heterotopic ossification (HO), which is also the focus of this paper. The Hatsell’2015 reference shows that it is Activin A that matters, as its pharmacologic inhibition using an anti-Activin A monoclonal antibody completely blocks HO. Furthermore, follow-on work from the same group shows that even developing HO lesion can be stopped and even partially regressed when Activin A is inhibited (see PMID 28782882).

c. These results have been independently replicated by Lees-Sheppard et al (PMID 29396429 and 30226468).

Therefore, there is overwhelming evidence that what drives HO in FOP is not “enhanced BMP signaling” but rather activation of BMP signaling via Activin A. The authors should revise the corresponding text to reflect this “new” (well, almost four years old now) understanding of how heterotopic bone forms in FOP. A repeat of prior interpretations that have been proven incorrect does not help the field advance, and merely lends more credence to what we now know is incorrect.”

We modified the introduction to refrain from pointing to increased BMP signaling and responsiveness to BMP ligands as the cause of FOP.

Point #2. and 3.1. “Along the same lines, the statement “whereas activin A normally transduces through the ACVR2-ACVR1B complex and inhibits Smad1/5/8 phosphorylation” is very confusing. Signaling by Activin A through the ACVR2-ACVR1B complex does not inhibit Smad1/5/8 phosphorylation directly – no evidence is being presented to demonstrate such an effect. (For the record, it has already been shown that signaling by Activin A via FOPmutant ACVR1 does not appear to affect signaling via ACVR1B, at least in the settings thus far tested – see Hatsell’2015.) What Activin A does is to form a non-signaling complex with ACVR1 (ALK2) in the context of its type II receptors. Although it is tempting to think of this interaction as an inhibitory one (as this is how the existence of the non-signaling complex has been demonstrated in the literature), perhaps the better way to think about this is that Activin A renders ACVR2A, ACVR2B, and ACVR1B unavailable to interact with other BMPs. This is a fine point, but one that should be clear to those that have followed the work published by the Elowitz lab (PMID28886385).

The authors should revise this statement for clarity, simply stating that while Activin A will transduce a pSmad2/3 signal through the ACVR2-ACVR1B complex, it forms a nonsignaling complex with ACVR2-ACVR1 (as long as ACVR1 is wild type). This complex is reinterpreted in FOP and perceived as a signaling complex, much like that which would result by ligation of these receptors by the BMPs that can interact with ACVR1.”

Also along these lines, the statement “Similar to FOP, non-genetically-driven HO also arises from excessive BMP signaling...” is erroneous, simply because FOP does not arise from “excessive BMP signaling”. This should be clear by now. The authors should revise this statement to state something to the effect of “In contrast to FOP which results from improper activation of FOP-mutant ACVR1 by Activin A, non-genetically driven HO appears to arise from excessive BMP signaling...”.

Although the final signaling outcome is most likely the same (as both types of HO follow an endochondral path), it should be made clear that the initiating signal is very different! (What the two

types of HO have in common is that the effect is localized, and that it affects certain types of connective tissue, and all connective tissue or other organs.)

As suggested by the reviewer, we modified accordingly these sentences in the Introduction to clarify these concepts.

Point #3.2. “In the Results section, the authors review the origin of HO in FOP and claim that “osteochondroprogenitors have numerous origins”. This is a misreading of the corresponding literature, as Dey et al make it very clear in their cited manuscript that the cells that they have been studying are no other than FAPs. This quote from that article demonstrates this point: “Our observations that Mx1+ Lin–Sca1+PDGFRa+ interstitial cells exhibit markedly enhanced chondrogenic potential and decreased adipogenic potential with mutant ACVR1 and that, in muscle engraftment studies, the default adipogenic fate of WT Mx1+ interstitial cells was modified by mutant ACVR1 to form bone led us to postulate that Mx1+ HO progenitors represent “reprogrammed” fibroadipogenic progenitor cells.” Furthermore, at least in skeletal muscle, there is no evidence that there are “numerous” different types of progenitors; rather there are only two that have been described to date: the satellite cells (that give rise to myofibers when muscle grows or regenerates), and the FAPs (which participate in the repair process, but themselves do not give rise to myofibers). No other progenitors have been described, and hence the adjective “numerous” cannot stand. Based on the above, the authors should revise the corresponding statement aiming for clarity by simply contributing the cellular origin of HO to the activated FAPs.”

We modified accordingly this sentence to point to fibro/adipogenic precursors as the major cells of origin of heterotopic ossification.

Point #3.3 and 3.4. “In the Results section, p. 5, the authors claim that “A66, a PI3K α -specific inhibitor, also reduced SMAD1/5 levels in MSCs, decreased GSK3 phosphorylation, and reduced BMP responsiveness (Figure EV1B)”. This is experiment has its origin in their prior work (Gamez et al, 2016). However, the data shown in Figure EV1B does not support the conclusions reached here, except perhaps for pGSK3, and even there, the data is not consistent (i.e. not the same effect is seen between the left, middle, and right panels at the no added BMP point, but with exposure to A66). For Smad1 levels, there is no discernible difference between the A66-treated compared to untreated samples, and much the same can be said for the effect on pSmad1/5. The lack of consistency between these results and the prior publication (Gamez et al, 2016) is perplexing. Nonetheless, the results shown here do not support the claims in the text. Therefore, the text has to be either revised accordingly, or the experiments need to be repeated with the aim of getting clearer data.

4. Mirroring the previous comment, the data shown on Figure 1C although somewhat more consistent also does not show a convincing effect for the levels of SMAD1 protein or for that matter the degree of SMAD1/5 phosphorylation. Therefore, either the statement “A66 reduced SMAD1 protein levels and SMAD1/5 phosphorylation” (p.5) has to be revised, or the experiments have to be repeated and convincing data presented.”

We repeated some experiments and improved the quality of the western blots. New western blots are included in the Figures 1C and EV1B. Moreover, we also analyzed the effects of BYL719 on SMAD1 levels and its phosphorylation upon BMP6 or Activin A addition (Figure EV3B). Altogether, the data confirms that PI3K α inhibitors reduce SMAD1 levels and its phosphorylation upon ligand activation. In addition, PI3K α inhibitors also reduce pGSK3 and pS6 levels, suggesting inhibition of AKT and mTOR/S6K signaling.

Point #3.5. “The data pertaining to ACVR1[Q207D] shown in Figure 2 is confusing. This is a constitutively active variant that is not responsive to ligand. The responsiveness shown is likely due to other type I receptors being activated, particularly so with BMP2 which preferentially binds to ALK3 and ALK6 rather than ACVR1. In order to attribute an effect on Q207D by any BMP, the authors ought to employ a gate keeper strategy that isolates Smad1/5/8 signaling to the Q207D variant only, or they should notify the reader about the possibility that what is being read out is a combination of signaling from Q207D plus other type I BMP receptors.”

ACVR1 GS domain mutants (including R206H and Q207D), are predicted to alter the α -helical structure of the GS domain, disrupting interactions with FKBP12 and unestabilizing the inactive configuration (Chaikuad et al., 2012: PMID:22977237). Therefore, the Q207D mutation renders ACVR1 constitutively active yet, under certain conditions, ACVR1 Q207D is hypersensitive to ligand stimulation (e.g. Bagarova et al., 2013: PMID:23572558; Haupt et al., 2018: PMID:29097342). Therefore, expression of ACVR1Q207D in different cellular contexts leads to a

constitutive basal activity that could be further enhanced upon addition of ligands. Interestingly, ablation of BMPR2 and ACVR2A abrogated constitutive activation and ligand-mediated signaling of ACVR1Q207D and disrupted heterotopic ossification in ACVR1Q207D mice (Bagarova et al., 2013: PMID:23572558). Therefore, signal from the Q207D mutant requires at least the scaffolding function of type II receptors. In addition, it has been shown a BMP2 and BMP6 ligand-induced heterodimerization between ACVR1 and BMPRIA (ALK3) (Traeger et al., 2018: PMID:30227271). Altogether, this evidence suggest the formation of a heteromeric receptor between ACVR1Q207D and endogenous type I and type II receptors that would depend on the relative expression levels of the receptors and their ligand affinities. As the reviewer indicate, that would be especially relevant for BMP2 that has much higher affinities for BMPRIA and BMPRIB. We think that the identification of receptor complexes between ACVR1Q207D and specific endogenous receptors in MSCs is beyond the scope of our manuscript.

Point #3.6 to 3.8. “In the Discussion section, the statement “We have identified that PI3K α inhibitors (A66 or BYL719) block several BMP-activated signaling pathways required for HO in progenitor cells” is a misinterpretation of the actual findings. It is true that these inhibitors appear to affect several different players that are activated by BMPs: SMAD1 (and presumably also 5?), GSK3, p38, S6. But this does not mean that these “...pathways are required for HO in progenitor cells”. In other words, it is possible that inhibition of Smad1/5/8 signaling alone would suffice. To demonstrate that any given player from the list of those affected by A66 or BYL719 (SMAD1, GSK3, p38, S6) is required for HO, other experiments are needed, such as genetic ablation or RNAi-mediated knockdown of their levels, presumably in the cells that matter (i.e. the FAPs).

7. In the Discussion section, the statement that “activation of mTOR... is also critical for FOP...” needs to be put into perspective with the fact that there are case reports of FOP patients that have been treated chronically with rapamycin but with no documented effect on their HO (PMID 29241828).

8. In the Discussion section, the authors justify the use of the Q207D model by claiming that “...it helped us to increase the penetrance and stringency of the phenotype...”. This is misleading. The two existing genetically correct models of FOP, where HO is driven using conditional knock-ins of Acvr1[R206H], present with very robust HO and have been used to discover and validate potential therapies. This is demonstrated in several publications (see PMID 26333933, 29396429, 28782882 and 30226468). It is incorrect to give the impression that the Q207D model is superior to those described in the listed publications, as it is not! This needs to be rectified accordingly.”

As suggested by the reviewer, we modified accordingly these sentences in the Discussion section to clarify these concepts. We refrain from claiming that all the pathways inhibited by PI3K α inhibitors downstream BMP receptors (SMAD1/5, mTOR, AKT, and GSK3) are required for HO. We also specify that mTOR has been shown to be critical for FOP only in preclinical mouse models. Finally, we avoid any comparison of penetrance and stringency between different mouse models of HO.

2nd Editorial Decision

1 July 2019

Thank you for the submission of your revised manuscript to EMBO Molecular Medicine. We have now received the enclosed report from the referee who was asked to re-assess it. As you will see the reviewer is now globally supportive and I am pleased to inform you that we will be able to accept your manuscript pending minor editorial amendments.

***** Reviewer's comments *****

Referee #2 (Comments on Novelty/Model System for Author):

The authors do not provide especially convincing mechanistic data to support the direct and specific effects of PI3K inhibition as a viable therapeutic for treatment of heterotopic ossification.

Referee #2 (Remarks for Author):

General comment:

The authors have done an adequate job at revising the MS to address reviewer concerns. However, what is now lacking is a discussion of why inhibiting PI3K would be preferable to the other therapies under investigation.

Specific comment:

The variability in Smad1 levels shown in the westerns is hard to understand as Smad1 should be a constant and the ratio of pSmad1 to total Smad1 an indication of signaling. As Smad1 levels seem to change, it is hard to figure out if signaling through Smad1 actually changed. The authors do not discuss this issue.

2nd Revision - authors' response

8 July 2019

Reviewer # 2

Point #1. "The authors have done an adequate job at revising the MS to address reviewer concerns. However, what is now lacking is a discussion of why inhibiting PI3K would be preferable to the other therapies under investigation."

There are advantages for implementing BYL719 as a therapeutic candidate for HO and FOP: First, inhibition of PI3K α in progenitors suppresses at least two signaling pathways required for progression of HO downstream of BMP receptors, SMAD1, and mTOR. Moreover, data indicate some capacity of cells without expression of the mutated allele to contribute to heterotopic lesions (PMID: 27881824; PMID: 29396429). Similarly, non-genetic HO develops in the absence of receptor alterations but with enhanced SMAD and mTOR activities (PMID: 29029772; PMID: 28716575). In any case, both mutated (ACVR1R206H) and wild type cells will be equally targeted by PI3K α inhibitors. In addition, BYL719 has already demonstrated a good tolerability and safety profile in humans. This evidence resulted in its approval last month (May 24th, 2019) by the FDA for treatment of PIK3CA-mutant advanced or metastatic breast cancer. This information has been included in the Discussion section.

Point #2. "The variability in Smad1 levels shown in the westerns is hard to understand as Smad1 should be a constant and the ratio of pSmad1 to total Smad1 an indication of signaling. As Smad1 levels seem to change, it is hard to figure out if signaling through Smad1 actually changed. The authors do not discuss this issue."

Our data confirm that PI3K α inhibitors reduce SMAD1 levels and its phosphorylation upon ligand activation. In addition, our results also confirm that addition of BMP2 increase the levels of SMAD1 as it was previously described (PMID:26896753). SMAD1 is known to be regulated by phosphorylation at several sites by different kinases (PMID:19114991). Most important events are the receptor-mediated C-terminal phosphorylation and GSK3/MAPK phosphorylation of the linker region of SMAD1 (PMID:18045539). These phosphorylations strongly modulate SMAD1 protein stability by the differential recognition by E3-ubiquitin ligases and nucleo-cytoplasmic shuttling of SMAD complexes (PMID:19114991; PMID: 21308777; PMID:11703946). This evidence emphasizes the tight control of SMAD1 activity not only by phosphorylation by BMP receptors, but also by the regulation of its steady-state protein levels by ubiquitin-proteasome degradation.

Corresponding Author Name: Francesc Ventura
Journal Submitted to: EMBO Molecular Medicine
Manuscript Number: EMM-2019-10567